# Antibiotic usage practices and its drivers in commercial chicken production in Bangladesh

**Sukanta Chowdhury**[1]*, **Guillaume Fournié**[2], **Damer Blake**[2], **Joerg Henning**[3], **Patricia Conway**[4], **Md. Ahasanul Hoque**[5], **Sumon Ghosh**[1], **Shahana Parveen**[1], **Paritosh Kumar Biswas**[5], **Zubair Akhtar**[1,6], **Khaleda Islam**[7], **Md. Ariful Islam**[1], **Md. Mahbubur Rashid**[1,8], **Ludvic Pelligand**[2], **Zobaidul Haque Khan**[7], **Mahmudur Rahman**[9], **Fiona Tomley**[2], **Nitish Debnath**[7], **Fahmida Chowdhury**[1]

1 International Centre for Diarrhoeal Disease Research (icddr,b), Dhaka, Bangladesh, 2 Royal Veterinary College, London, United Kingdom, 3 The University of Queensland, Brisbane, Australia, 4 Nanyang Technological University, Singapore, Singapore, 5 Chattogram Veterinary and Animal Sciences University, Chittagong, Bangladesh, 6 Biosecurity Program, The Kirby Institute, University of New South Wales (UNSW), Sydney, Australia, 7 Fleming Fund Country Grant to Bangladesh, DAI Global, LLC, Dhaka, Bangladesh, 8 School of Women's and Children's Health, University of New South Wales (UNSW), Sydney, Australia, 9 Global Health Development, EMPHNET, Dhaka, Bangladesh

* sukanta@icddrb.org

**Data Availability Statement:** All relevant data are within the paper and its Supporting Information files.

## Abstract

Irrational and inappropriate use of antibiotics in commercial chicken production can contribute to the development of antimicrobial resistance. We aimed to assess antibiotic usage in commercial chicken production in Bangladesh, and identify factors associated with this practice. We conducted a large-scale cross-sectional study to collect information on antibiotic usage in commercial chickens from January to May 2021. Structured interviews were conducted with 288 broiler, 288 layer and 192 Sonali (locally-produced cross-bred) farmers in 20 sub-districts across Bangladesh. The frequency of antibiotic usage, the types of antibiotics and purpose of usage were estimated for each production type. Adjusted odds ratios (aOR) were calculated to measure the association between antibiotic usage and factors related to the characteristics of the farms and farmers using multivariable logistic regression models. The proportion of farms, irrespective of their production type, reporting usage of antibiotics in the 24 hours preceding the interview was 41% (n = 314, 95% CI: 37–44%). Forty-five percent (n = 344, 41–48%) reported antibiotic usage in the last 72 hours, 86% (n = 658, 83–88%) in the last 14 days, and almost all farms, 98% (n = 753, 97–99%), had used antibiotics since the start of their production cycle. Use of antibiotics in the 24 hours preceding an interview was more frequently reported in broiler (OR 1.91, 95% CI: 1.36–2.69) and Sonali (OR 1.94, 95% CI: 1.33–2.33) than layer farms. Oxytetracycline (23–31%, depending on production type), doxycycline (18–25%), ciprofloxacin (16–26%) and amoxicillin (16–44%) were the most frequently used antibiotics. Antibiotics were reported to be used for both treatment and prophylactic purposes on most farms (57–67%). Usage of antibiotics in the 24h preceding an interview was significantly associated with the occurrence of any illnesses in chickens (aOR broiler: 41.22 [95% CI:13.63–124.62], layer: aOR 36.45[9.52–139.43], Sonali: aOR 28.47[4.97–162.97]). Antibiotic usage was mainly advised by veterinary practitioners (45–71%, depending on production type), followed by feed dealers (21–

**Funding:** This study was funded by the Fleming Fund Country Grant to Bangladesh (FF48-416FFCGB1). The funders had no role in study design, data collection and analysis, decision to publish, or preparation of the manuscript.

**Competing interests:** The authors have declared that no competing interests exist.

40%) and farmers (7–13%). Improvement of chicken health through good farming practices along with changes in key stakeholders (feed dealers and practitioners) attitudes towards antibiotic recommendations to farmers, may help to reduce the levels of antibiotic usage and thus contribute to mitigate antimicrobial resistance.

## Introduction

Globally, antimicrobial resistance (AMR) is one of the greatest threats to public health [1]. Inappropriate use of antimicrobials in humans, poultry, fish, and livestock has contributed to AMR emergence [2,3]. Antimicrobials are frequently used as prophylactic drugs in commercial animal production systems in low-and middle-income countries [4–9]. A study suggests that global consumption of antimicrobials in animals will rise by 67% by 2030 [10]. According to a review study, antimicrobial consumption in animals is threefold that of human consumption [11]. Despite the benefit of treating animal diseases using antimicrobial drugs, the development of AMR in both animal and human associated bacterial populations has raised global concerns [12].

In Bangladesh, commercial chicken production is greatly expanding to meet the rising demand for meat and eggs for human consumption. The commercial chicken industry includes broiler, layer and Sonali intensive farms. Broilers and layers are exotic chickens reared for meat and eggs, respectively. Sonali is a locally-produced cross-bred between Rhode Island Red male and Fayoumi female, reared for both meat and egg production. Sonali chickens are usually slaughtered for meat from 12 weeks to at the end of laying period [13,14]. Multiple studies reported the evidence of antibiotic use in commercial broiler and layer chicken. A cross-sectional study in Bangladesh observed that 98% of commercial chicken farms used antimicrobials in the current production cycle and 85% of farmers administered antimicrobials for prophylactic purpose [15]. Another study reported similar findings where 100% of broiler farms used at least one antibiotic over the production cycle and 32% of the farms used antibiotics for prophylactic purpose [16]. In 2010, the Bangladesh government passed the "Bangladesh Fish Feed and Animal Feed Act 2010" banning the introduction of antibiotics, growth hormone, steroid and insecticides into animal feed during manufacturing [17]. Farmers usually administered antimicrobials to the chicken through water and feed [15]. Yet, the easy access and availability of over-the-counter antibiotics at feed dealer shops and pharmacies can play an important role for the emergence of AMR in Bangladesh [18]. A large number of animal feed dealers and drug sellers advise farmers to use antimicrobials for chicken production, despite limited understanding about the impact of excessive and prophylactic use of antibiotics on AMR emergence [19]. To date, Bangladesh has no drug policy or guideline for appropriate use of antibiotic to treat animals, except for the above-mentioned ban on the addition of antimicrobials during feed manufacture. Although multiple small studies were conducted to assess the antimicrobial usage in broiler and layer chicken, the extent of antibiotic usage in all major commercial chicken production types from wider regions and its drivers are not well explored. Increasing information on current practices related to antibiotic usage in commercial chicken production is crucial for the design of more effective interventions to minimize the animal and public health impact of AMR. To address this information gap, we conducted a cross-sectional study in intensive commercial exotic chicken production areas wider geographical location in Bangladesh to collect information about the nature of antibiotic usage and its drivers in commercial broiler, layer and Sonali chicken farms.

## Materials and methods

### Study design

A cross-sectional study was carried out to collect antibiotic usage information in commercial chicken farms in Bangladesh from January to May 2021. Over the study period, every selected farm was visited once to collect data. Farms were stratified based on their (i) location (upazilas, i.e. sub-district), (ii) production type (broiler, layer or Sonali), and (iii) scale (small: ≤1000 chickens, medium: 1001–2000, large: >2000 chickens).

Upazilas (sub-districts) with the highest number of commercial chicken farms were chosen from the selected district. Gazipur, Chattogram, and Cumilla districts were selected for broiler and layer farms; and Joypurhat and Bogura districts for Sonali farms (Fig 1) [14]. We chose Gazipur, Chattogram, and Cumilla districts because of higher broiler and layer farm density compared to other districts in Bangladesh. Similarly, we choose Joypurhat and Bogura districts because of higher Sonali chicken farm density. Through consultation with local livestock officers from the Bangladesh Department of Livestock Services (DLS) and feed dealers, the 4 upazilas with the highest number of farms in each district were then selected. In each of these selected upazilas, the field team enrolled 8 small, 8 medium and 8 large farms for each targeted production type (broiler, layer and/or Sonali depending on the upazila). In the absence of a reliable list of commercial farms, a snow ball sampling approach was used to identify targeted number of farms in each upazila. Feed dealers operating in the targeted upazilas were asked about the farms they do business with. A first farm was then recruited, and the farmer asked about the address of the nearest farm, which was then recruited. This procedure was repeated until reaching the targeted sample size for each production type and farm size category. Chicken flocks younger than 14 days were not included in this study, as we asked about antibiotic usage on the 14 days preceding the farm visit. Overall, 288 broiler and 288 layer farms were enrolled in 12 upazilas in 3 districts, and 192 Sonali farms were enrolled in 8 upazilas in 2 districts. Based on the previous study findings [16], we expected to estimate 32% prevalence of antibiotic usage in commercial chicken farms. The required number of sample size was 768 assuming a 32% expected prevalence, with 95% confidence interval level and 5% precision.

A total of 20 animal feed dealers were interviewed from the 20 above-mentioned selected upazilas. In each upazila, a large feed dealer was selected purposively who used to sell larger amount of poultry feed and medicine. The field team visited each selected feed dealer once (preferably during the busiest hours, as farmers visited their shop) to record drug dispensing practices by observing interactions with five consecutive chicken farmers. A total of 100 (five farmers per feed dealer) interactions with farmers were thus observed. Antibiotics available for sale in each shop were also recorded.

### Data collection

A structured questionnaire (supplement 1) was used to collect data from selected farms. Before data collection, written informed consent was obtained from all selected farmers, animal feed and chick dealers to participate in the study. We collected data on farm demographics (number of chickens at the time of the visit, number of poultry species, number of poultry sheds and poultry density), production stage, antibiotic usage in the 24 hours, 72 hours and 14 days preceding the interview, antibiotic usage since the start of the production cycle, name of antibiotics used, pro-biotic usage, purpose(s) of antibiotic usage, antibiotic prescribing practices by authorized practitioners, chicken morbidity during the day of the farm visit, chicken mortality over the last14 days, farmer's education, duration of farming experience and familiarity with the term "AMR". The questionnaire (supplement 2) for animal feed dealers covered antibiotic

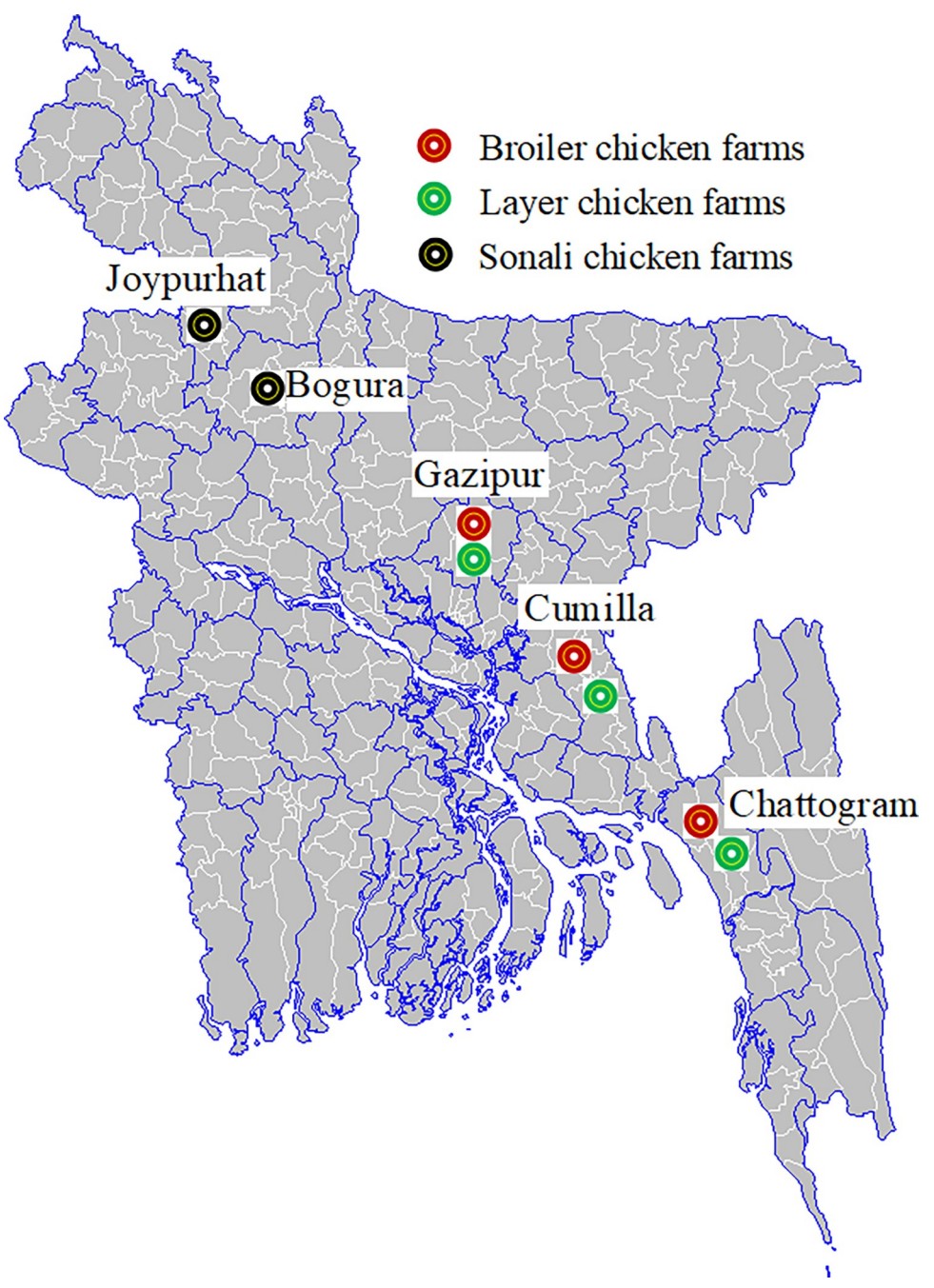

**Fig 1. Map of the study sites for commercial chicken farms sampling.**

dispensing practices, available antibiotics kept for sale, antibiotic prescribing practices by authorized practitioners and familiarity with the term "AMR".

## Statistical analysis

We summarized the characteristics of chicken farms, including flock size and production type, using descriptive analyses (frequency, mean, standard deviation, $p$-value and 95% confidence interval). Fisher's exact test was performed to examine the differences of proportion of

demographic characteristics between categories of each categorical variable (farm locations, number of batches of chicken, number of poultry sheds, sources of chicken feed, collection of day-old chicks). One-way ANOVA was conducted to assess the mean difference of health status, flock and farm size between categories of each categorical variable. The proportion of farms using each type of antibiotic and its 95% confidence interval was estimated separately for each production type. To describe the association between categorical farm management or demographic or geographic variables, and the use of antibiotics on chicken farms, we firstly estimated the odds ratio (OR) using bivariable logistic regression analysis. Then, we performed multivariable analyses to estimate adjusted odds ratio (aOR) for each production type using additional explanatory variables. Variables with a *p* value ≤0.2 at a likelihood ratio test were used to build a multivariable logistic regression model as described [20]. We used backward stepwise selection of variables with an inclusion threshold of 0.05. We used Hosmer-Lemeshow test to calculate model $\chi^2$ statistic and McFadden's pseudo-$R^2$ (the coefficient of determination) to explain variance and measure goodness-of-fit for multivariate regression model. All statistical analyses were performed in Stata 13 software (StataCorp LP, College Station, TX).

### Ethics statement

The study protocol was reviewed and approved by icddr,b Research Review Committee, Ethical Review Committee and Animal Experimentation Ethics Committee (PR-20116).

## Results

### Demographic characteristics of commercial chicken farms

Among the 768 surveyed commercial chicken farms, 688 farms (90%, 95% CI: 87–92%) only raised one production type of chicken (either broiler or layer or Sonali), and 9–12% of farms raised different types of chicken production. About half of farms (44–61%) had a single poultry shed. The average number of broiler chicken for small, medium and large farms was 782, 1415 and 2975, respectively. The average number of layer chicken for small, medium and large farms was 831, 1695 and 5432, respectively. The average number of Sonali chicken for small, medium and large farms was 893, 1787 and 6152, respectively. Most farms (69–92%) had a single batch of chickens during our visit. Almost all farms (93–100%) collected feed and most (51–75%) collected day-old chicks from feed and/or chick dealers. Others prepared feed at their farms (5–7%) and collected day-old chicks from hatcheries (25–49%). During the farm visits, the average age of broiler, layer and Sonali chickens was 22 days (95% CI: 21–23), 322 days (95% CI: 302–341) and 91 days (95% CI: 77–105), respectively (Table 1).

### Frequency and characteristics of antibiotic usage within the last 24 hours

Among the 768 commercial chicken farms, 41% (n = 314, 95% CI: 37–44%) reported having used antibiotics within the 24 hours preceding our visit. This proportion was higher among broiler (47%, 95% CI: 41–52%) and Sonali (47%, 95% CI: 40–54%) than in layer farms (31%, 95% CI: 26–37%) (Table 2). Two-third of farms (66–67% depending on production type) administered antibiotics for both therapeutic and prophylactic purposes. The proportion of farms that used antibiotic prophylactically and not therapeutically was slightly higher in layer (14%, 95% CI: 8–24%) than broiler (8%, 95% CI: 4–14%) and Sonali (3%, 95% CI: 1–9%) farms. Commercial chicken farms used diverse classes of antibiotics including tetracyclines, fluoroquinolones, macrolides, aminoglycosides, penicillins, and polymyxins. Doxycycline, oxytetracycline and ciprofloxacin were the most commonly reported antibiotics in broiler and layer chickens, whereas amoxicillin usage was more commonly reported in Sonali chickens

**Table 1. Demographic and health characteristics of commercial broiler (n = 288), layer (n = 288) and Sonali (n = 192) chicken farms sampled during January to May 2021 in Bangladesh.**

| Characteristics | Broiler farms | | | | Layer farms | | | | Sonali farms | | | |
|---|---|---|---|---|---|---|---|---|---|---|---|---|
| Areas, n (%) | Small | Medium | Large | Overall p | Small | Medium | Large | Overall p | Small | Medium | Large | Overall p |
| Gazipur | 32 (11) | 32 (11) | 32 (11) | 1 | 32 (11) | 32 (11) | 32 (11) | 1 | - | - | - | |
| Chattogram | 32 (11) | 32 (11) | 32 (11) | | 32 (11) | 32 (11) | 32 (11) | | - | - | - | 1 |
| Cumilla | 32 (11) | 32 (11) | 32 (11) | | 32 (11) | 32 (11) | 32 (11) | | - | - | - | |
| Joypurhat | - | - | - | | - | - | - | | 32 (17) | 32 (17) | 32 (17) | |
| Bogura | - | - | - | | - | - | - | | 32 (17) | 32 (17) | 32 (17) | |
| **Number of batches of chicken, n (%)** | | | | | | | | | | | | |
| Single | 96 (33) | 91 (32) | 79 (27) | <0.001 | 87 (30) | 76 (26) | 35 (12) | <0.001 | 63 (33) | 57 (30) | 32 (17) | <0.001 |
| Two | - | 5 (2) | 15 (5) | | 9 (3) | 19 (7) | 46 (16) | | 1 (1) | 7 (4) | 18 (9) | |
| Three or more | - | - | 2 (1) | | - | 1 (1) | 15 (5) | | - | - | 14 (7) | |
| **Number of poultry sheds, n (%)** | | | | | | | | | | | | |
| Single | 74 (24) | 60 (21) | 42 (15) | | 60 (21) | 42 (15) | 25 (7) | | 48 (25) | 33 (17) | 6 (3) | |
| Two | 20 (7) | 30 (10) | 35 (12) | <0.001 | 27 (9) | 34 (12) | 23 (8) | <0.001 | 12 (4) | 28 (10) | 22 (8) | <0.001 |
| More than two | 2 (1) | 6 (2) | 19 (7) | | 9 (3) | 20 (7) | 48 (17) | | 4 (1) | 3 (1) | 36 (13) | |
| **Source of chicken feed, n (%)** | | | | | | | | | | | | |
| Feed dealers | 96 (33) | 96 (33) | 96 (33) | | 94 (33) | 91 (32) | 87 (30) | | 62 (32) | 62 (32) | 55 (29) | |
| Home made | - | - | - | undefined | 2 (1) | 5 (2) | 8 (3) | 0.129 | 2 (1) | 2 (1) | 9 (5) | 0.03 |
| Both sources | - | - | - | | - | - | 1 (1) | | - | - | - | |
| **Collection of day-old chicks, n (%)** | | | | | | | | | | | | |
| Hatchery | 21 (7) | 19 (7) | 31 (11) | | 41 (14) | 46 (16) | 54 (19) | | 23 (12) | 18 (9) | 30 (16) | |
| Feed and/or chick dealer | 75 (26) | 77 (27) | 65 (23) | 0.108 | 55 (19) | 50 (17) | 42 (15) | 0.177 | 41 (21) | 46 (24) | 34 (18) | 0.093 |
| **Health status of chicken flock, mean (standard deviation)** | | | | | | | | | | | | |
| Average number of sick chickens within last 14 days | 65 (117) | 88 (124) | 178 (226) | <0.001 | 68 (24) | 66 (22) | 77 (27) | 0.242 | 48 (25) | 55 (29) | 60 (31) | 0.245 |
| Average number of dead chickens within last 14 days | 36 (62) | 46 (65) | 104 (120) | <0.001 | 61 (21) | 59 (20) | 73 (25) | 0.057 | 48 (25) | 55 (29) | 60 (31) | 0.486 |
| **Average number of chickens/farm, mean (standard deviation)** | 782 (215) | 1451 (293) | 2975 (1289) | <0.001 | 831 | 1695 | 5432 | <0.001 | 893 | 1787 | 6152 | <0.001 |
| **Average size of farm shed in Sq-feet, mean (standard deviation)** | 1102 (601) | 1618 (689) | 3170 (6035) | <0.001 | 1305 | 2007 | 3349 | 0.002 | 1415 | 2100 | 2821 | 0.018 |
| **Average age of the chicken in days, mean (standard deviation)** | 22 (6) | 23 (9) | 22 (5) | 0.557 | 335 | 295 | 335 | 0.698 | 87 | 63 | 123 | 0.710 |

Statistically significant differences are indicated within each characteristic.

(Table 3). According to reports from farmers, antibiotics were mostly recommended by veterinary practitioners (55%), followed by poultry feed dealers (30%) and the farmers themselves (9%). Broiler farmers relied more frequently on feed dealer's advice about antibiotic usage than layer and Sonali chicken farmers (Table 2).

**Table 2. Farm-level antibiotic use within the 24 hours preceding visits to commercial broiler, layer and Sonali chicken farms in Bangladesh during January-May 2021.**

| Variables | Broiler(N = 288) | | Layer (N = 288) | | Sonali (N = 192) | |
|---|---|---|---|---|---|---|
| | Number of farms n (%) | 95% CI | Number of farms n (%) | 95% CI | Number of farms n (%) | 95% CI |
| **Uses of at least one antibiotic** | 134 (47) | 41–52 | 90 (31) | 26–37 | 90 (47) | 40–54 |
| **Number of antibiotics** | **N = 134** | | **N = 90** | | **N = 90** | |
| Single antibiotic | 74 (55) | 46–63 | 44 (49) | 38–59 | 49 (54) | 44–65 |
| Two antibiotics | 48 (36) | 27–44 | 34 (38) | 27–48 | 36 (40) | 30–51 |
| Three or more antibiotics | 12 (9) | 4–15 | 12 (13) | 7–22 | 5 (6) | 2–12 |
| **Purposes of antibiotic use** | **N = 134** | | **N = 90** | | **N = 90** | |
| Prophylaxis | 11 (8) | 4–14 | 13 (14) | 8–24 | 3 (3) | 1–9 |
| Treatment | 34 (25) | 18–33 | 18 (20) | 14–32 | 27 (30) | 21–41 |
| Both | 89 (66) | 57–74 | 59 (66) | 54–75 | 60 (67) | 56–76 |
| **Antibiotic suggested by** | **N = 134** | | **N = 90** | | **N = 90** | |
| Veterinary practitioner | 60 (45) | 36–53 | 49 (54) | 43–64 | 64 (71) | 61–80 |
| Pharmacy owner | 1 (1) | 1–4 | 3 (3) | 1–9 | - | - |
| Feed dealer | 54 (40) | 31–49 | 20 (22) | 14–32 | 19 (21) | 13–31 |
| Veterinary doctor from pharmaceutical company | 1 (1) | 1–4 | 1 (1) | 1–6 | 1 (1) | 1–6 |
| Quack | 1 (1) | 1–4 | 11 (12) | 6–20 | - | - |
| Self-decision | 17 (13) | 7–19 | 6 (7) | 2–13 | 6 (7) | 2–14 |

## Frequency and characteristics of antibiotic usage earlier in the production cycle

Almost all farms (98%, 95% CI: 97–99%, n = 753) had used antibiotics at least once between the starting date of the production cycle and date of our farm visit. In the 72 hours and 14 days

**Table 3. Types of antibiotics used within the 24 hours preceding visits to broiler (N = 134), layer (N = 90) and Sonali (N = 90) farms in Bangladesh during January-May 2021.**

| Name of the antibiotic | Number of broiler farms n (%) | 95% CI | Number of layer farms n (%) | 95% CI | Number of Sonali farms n (%) | 95% CI |
|---|---|---|---|---|---|---|
| **Tetracycline** | | | | | | |
| Doxycycline | 34 (25) | 18–34 | 27 (30) | 21–41 | 16 (18) | 11–27 |
| Oxytetracycline | 31 (23) | 16–31 | 28 (31) | 22–42 | 23 (26) | 17–36 |
| **Fluoroquinolones** | | | | | | |
| Ciprofloxacin | 30 (22) | 16–30 | 23 (26) | 17–36 | 14 (16) | 9–25 |
| Levofloxacin | 11 (8) | 4–14 | 7 (8) | 3–15 | 15 (17) | 10–26 |
| Enrofloxacin | 8 (6) | 3–11 | 3 (3) | 1–9 | 2 (2) | 1–8 |
| **Macrolides** | | | | | | |
| Erythromycin | 11 (8) | 4–14 | 7 (8) | 3–15 | 4 (4) | 1–11 |
| Azithromycin | 7 (5) | 2–10 | 4 (4) | 1–11 | - | - |
| Tylosin | 7 (5) | 2–10 | 2 (2) | 1–8 | 8 (9) | 4–17 |
| **Aminoglycosides** | | | | | | |
| Neomycin | 16 (12) | 7–19 | 9 (10) | 5–18 | 3 (3) | 1–9 |
| **Penicillins** | | | | | | |
| Amoxicillin | 21 (16) | 10–23 | 16 (18) | 11–27 | 40 (44) | 34–55 |
| **Polymyxins** | | | | | | |
| Colistin | 4 (3) | 1–7 | 8 (9) | 4–17 | 1 (1) | 1–6 |

**Table 4. Drugs, vaccines and associated products used within the 72 hours preceding visits to selected broiler (N = 288), layer (N = 288) and Sonali (N = 192) farms in Bangladesh during January-May 2021.**

| Product | Broiler farms | | Layer farms | | Sonali farms | |
|---|---|---|---|---|---|---|
| | Yes n (%) | No n (%) | Yes n (%) | No n (%) | Yes n (%) | No n (%) |
| Vitamins | 218 (76) | 70 (24) | 217 (75) | 71 (25) | 186 (97) | 6 (3) |
| Minerals | 52 (18) | 236 (82) | 56 (19) | 232 (81) | 70 (36) | 122 (64) |
| Antibiotics | 135 (47) | 153 (53) | 107 (37) | 181 (63) | 102 (53) | 90 (47) |
| Antifungal | 43 (15) | 245 (85) | 60 (21) | 228 (79) | 65 (34) | 127 (66) |
| Antiprotozoal | 19 (7) | 269 (93) | 20 (7) | 268 (93) | 41 (21) | 151 (79) |
| Anthelminthic | 13 (5) | 275 (95) | 31 (11) | 257 (89) | 12 (6) | 180 (94) |
| Growth promoters | 17 (6) | 271 (94) | 14 (5) | 274 (95) | 24 (12) | 168 (88) |
| Probiotics | 82 (28) | 206 (72) | 95 (33) | 193 (67) | 75 (39) | 117 (61) |
| Vaccines | 69 (24) | 219 (76) | 52 (18) | 236 (82) | 66 (34) | 126 (66) |

preceding the interview, 45% (n = 344, 95% CI 41–48%) and 86% (n = 658, 95% CI 83–88%) of farms reported having used antibiotics, respectively (Tables 4 and 5). The purposes of antibiotic usage in the last 14 days were similar to those reported for the last 24 hours, with most farmers (57–67% depending on production type) administering antibiotics for both prophylactic and therapeutic usage. The frequency of prophylactic usage of antibiotics only was comparatively higher in layer (16%) than broiler (8%) and Sonali (3%) farms. Likewise, similarly to antibiotics used in the last 24 hours, antibiotics used in the last 14 days were mostly prescribed by veterinary practitioners (59%), followed by poultry feed dealers (29%) and farmers themselves (7%). Broiler farmers relied more frequently on feed dealer's advice about antibiotic usage than layer and Sonali chicken farmers (Table 5).

## General practices with regards to antibiotic usage

More than 90% of commercial chicken farmers used antibiotics in their flocks for 3–7 consecutive days. According to the farmers' report, 55% broiler farms, 42% layer farms and 21% Sonali farms used antibiotics on the first day of the batch production cycle prophylactically.

**Table 5. Farm-level antibiotics used within the 14 days preceding visits to commercial broiler, layer and Sonali chicken farms in Bangladesh, during January-May 2021.**

| Variables | Broiler (N = 288) | | Layer (N = 288) | | Sonali (N = 192) | |
|---|---|---|---|---|---|---|
| | Number of farms n (%) | 95% CI | Number of farms n (%) | 95% CI | Number of farms n (%) | 95% CI |
| **Usage of at least one antibiotic** | 282 (98) | 96–99 | 207 (72) | 66–77 | 169 (88) | 83–92 |
| **Purposes of antibiotic use** | N = 282 | | N = 207 | | N = 169 | |
| Prophylaxis | 21 (8) | 5–11 | 34 (16) | 12–22 | 5 (3) | 1–7 |
| Treatment | 73 (26) | 21–31 | 45 (22) | 16–28 | 67 (40) | 32–47 |
| Both | 188 (67) | 61–72 | 128 (62) | 55–68 | 97 (57) | 50–65 |
| **Antibiotic suggested by** | N = 282 | | N = 207 | | N = 169 | |
| Veterinary practitioner | 147 (52) | 46–58 | 120 (58) | 51–65 | 120 (71) | 64–78 |
| Pharmacy owner | 1 (1) | 1–2 | 3 (1) | 1–4 | - | - |
| Feed dealer | 102 (36) | 31–42 | 47 (23) | 17–29 | 43 (25) | 19–33 |
| Veterinary doctor from pharmaceutical company | 2 (1) | 1–3 | 27 (13) | 9–18 | 2 (1) | 1–4 |
| Quack | 1 (1) | 1–2 | - | - | - | - |
| Self-decision | 29 (10) | 7–14 | 10 (5) | 2–9 | 4 (2) | 1–4 |

Many farmers (16–41%) reported using antibiotics for chicken fattening. While the majority (67–87%) of the farmers had heard about withdrawal periods for antibiotics, 3–5% of farmers planned to use antibiotics on the last day of the production cycle, just before sales. A few layer (n = 11, 4%) and Sonali farmers (n = 12, 6%) reported that they mixed antibiotics into the chicken feed for administration. There was no history of mixing antibiotic in chicken feed by broiler farmers. Most of the broiler (99%), layer (92%) and Sonali (100%) chicken farmers administered antibiotic to chicken through water. According to farmers' estimations, the mean cost of antibiotics per production cycle was 80 USD (standard deviation, SD ±58), 541 USD (SD ±472) and 172 USD (SD ± 151) for 1000 broiler, layer and Sonali chicken, respectively.

## Health status of commercial chicken flocks

On the day of farm visit, 103 broiler (36%), 58 layer (20%) and 69 Sonali (36%) farms had at least one sick chicken. Most surveyed farms reported that at least one chicken was sick (97% broiler,74% layer and 85% Sonali farms) and/or died (97% broiler, 67% layer and 85% Sonali farms) within the 14 days preceding our farm visit. According to the farmers' report, the overall proportion of sick chickens per farm within the 14 days preceding our farm visit was 71 per 1000 broiler, 21per 1000 layer and 31 per 1000 Sonali farms, whereas the overall proportion of dead chickens was 39per1000 broiler, 9 per 1000 layer and 23 per 1000 Sonali chickens.

## Interaction between farmers, feed dealers and other associated partners

Most farmers (80% broiler, 59% layer and 78% Sonali chicken farms) had interactions with feed dealers. Farmers received support from feed dealers mainly on feed supply, followed by chick supply, medicine supply, health care services with or without vets, sale of mature chickens and eggs, and the provision of credit. According to the farmers' reports, chicken production depended on credit from feed dealers (22–36% farms) and pre-existing agreements (other than credit including chick supply, feed supply and sale mature chicken) with feed dealers (25–41% farms). Layer farms were less dependent on such arrangement than broiler and Sonali farms (Table 6).

**Table 6. Nature of interaction between farmers, feed dealers and other associated partners.**

| Characteristics | Broiler farms n (%) | Layer farms n (%) | Sonali farms n (%) |
|---|---|---|---|
| Presence of interaction between farmers and feed dealers | 229 (80) | 166 (58) | 149 (78) |
| **Types of support provided by feed dealers** | | | |
| Feed supply to the farms | 225 (78) | 166 (58) | 142 (74) |
| Chick supply to the farms | 189 (66) | 116 (40) | 142 (74) |
| Medicine supply to the farms | 185 (64) | 137 (48) | 137 (71) |
| Selling chicken and eggs | 180 (63) | 87 (30) | 97 (51) |
| **Chicken production depends on** | | | |
| Credits from feed dealers | 103 (36) | 63 (22) | 63 (33) |
| Credits from large commercial farms | 28 (10) | 29 (10) | 35 (18) |
| Credits from hatcheries | 27 (10) | 9 (3) | 31 (16) |
| Agreements between farmers and feed dealers | 106 (37) | 72 (25) | 78 (41) |
| Agreements between farmers and large farms | 2 (1) | 2 (1) | 2 (1) |
| Agreements between farmer and hatcheries | 8 (3) | 17 (6) | 6 (3) |
| No dependency (no financial agreements) | 73 (25) | 143 (50) | 47 (24) |

### Antibiotic dispensing at feed dealers

During visits to the shops of feed dealers, vitamins (72%, n = 72) were most commonly dispensed to farmers, followed by antibiotics (41%, n = 41) and probiotics (30%, n = 30). According to our observation during feed dealers visit, broiler farmers (42%) mostly came to bought medicine followed by Sonali (40%) and layer (18%) farmers. Among the farmers purchasing antibiotics, most were advised by qualified veterinarians (n = 22, 54%), feed dealers (n = 2, 5%), or the decision was based on their own experience (n = 17, 41%). Six to twelve classes of antibiotics were available in visited feed dealers' shop. According to the self-reported data, 70% of feed dealers usually suggest 10–30% of farmers to buy antibiotics. Most feed dealers (85%) said they knew about AMR and all had knowledge on antibiotic withdrawal periods. Many feed dealers (60%) believed that antibiotics are mixed in poultry feed by commercial feed producers despite understanding that antibiotic use in commercial poultry feed is banned by the Bangladesh government.

### Factors associated with antibiotic usage in commercial chicken flocks in the 24 hours preceding the farm visit

The bivariable regression analysis showed that the overall usage of antibiotics in the previous 24 hours appeared to be higher in broiler (OR 1.91, 95% CI: 1.36–2.69) and Sonali (OR 1.94, 1.33–2.83) than layer farms. The odds of antibiotic usage were higher in medium (OR 1.41, 0.99–2.01) and large (OR 1.39, 0.97–1.98) than small farms (Table 7). The broiler (OR 2.79, 0.93–8.34) and layer (OR 3.09, 1.05–9.08) chicken farms located in Gazipur were more likely to use antibiotics than those in Cumilla (Tables 8 and 9). Sonali chicken farms located in Bogura (OR 3.67, 1.15–11.7) were more like to use antibiotics than those in Joypurhat (Table 10).

The multivariable regression analyses suggested that the occurrence of illness of any type (at least one sick chicken within the preceding 24 hours) was associated with higher odds of antibiotic usage in broiler (aOR 41.22, 13.63–124.62), layer (aOR 36.45, 9.52–139.43) and

**Table 7. Comparison of antibiotic use in commercial chicken farms (N = 768) by production type, flock size and areas, during January-May 2021, Bangladesh.**

| Characteristics | Antibiotic use within last 24 hours | | OR, 95% CI |
|---|---|---|---|
| | Yes | No | |
| **Chicken production type** | | | |
| Layer | 90 | 198 | Ref. |
| Broiler | 134 | 154 | 1.91 (1.36–2.69) |
| Sonali | 90 | 102 | 1.94 (1.33–2.83) |
| **Flock size** | | | |
| Small | 91 | 165 | Ref. |
| Medium | 112 | 144 | 1.41 (0.99–2.01) |
| Large | 111 | 145 | 1.39 (0.97–1.98) |
| **Areas for broiler and layer chicken** | | | |
| Cumilla | 55 | 137 | Ref. |
| Gazipur | 103 | 89 | 2.88 (1.89–4.39) |
| Chattogram | 66 | 126 | 1.3 (0.85–2) |
| **Areas for Sonali** | | | |
| Joypurhat | 30 | 66 | Ref. |
| Bogura | 60 | 36 | 3.67 (1.15–11.7) |

**Table 8. Factors associated with antibiotic use in commercial broiler farms (N = 288), during January-May 2021, Bangladesh.**

| Factors | Antibiotic use within last 24 hours | | OR, 95% CI | p | Adjusted OR, 95% CI | P |
|---|---|---|---|---|---|---|
| | Yes | No | | | | |
| **Farm categories** | | | | | | |
| Small | 42 | 54 | Ref. | | | |
| Medium | 46 | 50 | 1.18 (0.62–2.25) | 0.610 | | |
| Large | 46 | 50 | 1.18 (0.62–2.26) | 0.611 | | |
| **Study site** | | | | | | |
| Cumilla | 33 | 63 | Ref. | | Ref. | |
| Chattogram | 44 | 52 | 1.61 (0.5–5.22) | 0.423 | 1.12 (0.55–2.29) | 0.754 |
| Gazipur | 57 | 39 | 2.79 (0.93–8.34) | 0.066 | 2.33 (1.02–5.33) | 0.044 |
| **Age of the broiler chicken flock** | | | | | | |
| ≥31 days | 5 | 11 | Ref. | | | |
| 16–30 days | 103 | 118 | 1.92 | 0.367 | | |
| 1–15 days | 26 | 25 | 2.29 | 0.303 | | |
| **Presence of current illnesses in the chicken flock** | | | | | | |
| No | 40 | 145 | Ref. | | Ref. | |
| Yes | 94 | 9 | 37.86 (13.32–107.64) | <0.001 | 41.22 (13.63–124.62) | < 0.001 |
| **Received training on chicken production** | | | | | | |
| Yes | 30 | 58 | Ref. | | | |
| No | 104 | 96 | 2.09 (1.12–3.91) | 0.02 | | |
| **Knowledge on the purpose of antibiotic use** | | | | | | |
| Used to treat viral diseases | 5 | 12 | Ref. | | Ref. | |
| Used to treat bacterial diseases | 9 | 15 | 1.44 (0.49–4.62) | 0.540 | 2.73 (1.23–6.06) | 0.013 |
| Used to treat all diseases | 113 | 121 | 2.24 (0.78–6.43) | 0.133 | 3.25 (1.69–6.26) | <0.001 |
| Used to increase production | 7 | 6 | 2.8 (0.95–8.25) | 0.062 | 6.69 (2.8–16) | <0.001 |
| **Farmers education** | | | | | | |
| Graduate | 17 | 26 | Ref. | | | |
| Higher Secondary | 25 | 24 | 1.59 (0.82–3.09) | 0.168 | | |
| Secondary | 67 | 70 | 1.46 (0.74–2.89) | 0.273 | | |
| Primary | 20 | 31 | 0.99 (0.55–1.75) | 0.964 | | |
| Illiterate | 5 | 3 | 2.55 (0.48–13.5) | 0.271 | | |
| **Farming experiences** | | | | | | |
| > 10 years | 41 | 50 | Ref. | | | |
| 5–10 years | 31 | 53 | 0.71 (0.42–1.2) | 0.207 | | |
| 1–5 years | 47 | 37 | 1.55 (0.94–2.56) | 0.088 | | |
| < 1 year | 14 | 14 | 1.22 (0.45–3.27) | 0.694 | | |
| **Heard of AMR** | | | | | | |
| Yes | 78 | 107 | Ref. | | | |
| No | 56 | 47 | 1.63 (0.95–2.81) | 0.076 | | |
| **Antibiotic suggested by** | | | | | | |
| Veterinary doctor | 66 | 80 | Ref. | | | |
| Feed dealer | 61 | 61 | 1.21 (0.81–1.82) | 0.353 | | |
| Farmer | 7 | 13 | 0.65 (0.29–1.45) | 0.297 | | |

Model fit:model $\chi 2$ 20.81, $p$ 0.06 and $R^2$0.388.

**Table 9. Factors associated with antibiotic use in commercial layer farms (N = 288), during January-May 2021, Bangladesh.**

| Factors | Antibiotic use within last 24 hours | | OR, 95% CI | p | Adjusted OR, 95% CI | p |
|---|---|---|---|---|---|---|
| | Yes | No | | | | |
| **Farm categories** | | | | | | |
| Small | 23 | 73 | Ref. | | | |
| Medium | 32 | 64 | 1.58 (0.91–2.75) | 0.101 | | |
| Large | 35 | 61 | 1.82 (0.86–3.85) | 0.117 | | |
| **Study site** | | | | | | |
| Cumilla | 22 | 74 | Ref. | | Ref. | |
| Chattogram | 22 | 74 | 1 (0.32–3.1) | 1 | 1.08 (0.44–2.66) | 0.624 |
| Gazipur | 46 | 50 | 3.09 (1.05–9.08) | 0.040 | 3.79 (1.71–8.38) | <0.001 |
| **Age of the layer chicken flock** | | | | | | |
| ≥181 days | 62 | 155 | Ref. | | Ref. | |
| 91–180 days | 15 | 26 | 1.44 (0.69–3.01) | | 0.99 (0.45–2.24) | |
| 31–90 days | 9 | 11 | 2.04 (0.84–4.98) | | 2.98 (1.45–6.11) | |
| 16–30 days | 4 | 5 | 2 (0.53–7.51) | | 1.66 (0.34–8.16) | |
| 1–15 days | 0 | 1 | undefined | | undefined | |
| **Presence of current illnesses in the chicken flock** | | | | | | |
| No | 40 | 190 | Ref. | | Ref. | |
| Yes | 50 | 8 | 29.68 (9.38–93.92) | <0.001 | 36.45 (9.52–139.43) | <0.001 |
| **Received training on chicken production** | | | | | | |
| Yes | 31 | 60 | Ref. | | | |
| No | 59 | 138 | 0.82 (0.43–1.57) | 0.564 | | |
| **Knowledge on the purpose of antibiotic use** | | | | | | |
| Used to treat viral diseases | 4 | 10 | Ref. | | | |
| Used to treat bacterial diseases | 9 | 38 | 0.59 (0.27–1.28) | 0.183 | | |
| Used to treat all diseases | 71 | 145 | 1.22 (0.39–3.85) | 0.730 | | |
| Used to increase production | 6 | 5 | 3 (1.03–8.78) | 0.045 | | |
| **Farmers education** | | | | | | |
| Graduate | 12 | 41 | Ref. | | | |
| Higher Secondary | 22 | 24 | 2.21 (0.88–5.56) | 0.092 | | |
| Secondary | 48 | 107 | 1.53 (0.87–2.69) | 0.136 | | |
| Primary | 8 | 16 | 1.7 (0.6–4.84) | 0.314 | | |
| **Farming experiences** | | | | | | |
| > 10 years | 50 | 70 | Ref. | | | |
| 5–10 years | 20 | 64 | 0.44 (0.23–0.83) | 0.011 | | |
| 1–5 years | 17 | 57 | 0.42 (0.18–0.97) | 0.043 | | |
| < 1 year | 3 | 7 | 0.6 (0.24–1.5) | 0.275 | | |
| **Heard of AMR** | | | | | | |
| Yes | 54 | 141 | Ref. | | | |
| No | 36 | 57 | 1.65 (0.75–3.62) | 0.212 | | |
| **Antibiotic suggested by** | | | | | | |
| Veterinary doctor | 66 | 142 | Ref. | | | |
| Feed dealer | 20 | 49 | 0.88 (0.32–2.41) | 0.801 | | |
| Farmer | 4 | 7 | 1.23 (0.21–7.06) | 0.817 | | |

Model fit: Model $\chi^2$ 10.67, p 0.6387 and $R^2$ 0.3308.

**Table 10. Factors associated with antibiotic use in commercial Sonali farms (N = 192), during January-May 2021, Bangladesh.**

| Factors | Antibiotic use within last 24 hours | | OR, 95% CI | p | Adjusted OR, 95% CI | P |
|---|---|---|---|---|---|---|
| | Yes | No | | | | |
| **Farm categories** | | | | | | |
| Small | 26 | 38 | Ref. | | | |
| Medium | 34 | 30 | 1.66 (0.75–3.64) | 0.209 | | |
| Large | 30 | 34 | 1.29 (0.73–2.28) | 0.383 | | |
| **Study site** | | | | | | |
| Joypurhat | 30 | 66 | Ref. | | | |
| Bogura | 60 | 36 | 3.67 (1.15–11.7) | 0.028 | | |
| **Age of the Sonali chicken flock** | | | | | | |
| ≥181 days | 5 | 26 | Ref. | | Ref. | |
| 91–180 days | 7 | 18 | 2.02 (0.61–6.71) | | 3.67 (1.72–7.82) | 0.001 |
| 31–90 days | 31 | 34 | 4.74 (1.62–13.85) | | 7.09 (2.43–20.67) | <0.001 |
| 16–30 days | 41 | 22 | 9.69 (3.29–28.57) | | 12.76 (2.59–62.75) | 0.002 |
| 1–15 days | 6 | 2 | 15.6 (5.26–46.23) | | 34.75 (11.85–101.86) | <0.001 |
| **Presence of current illnesses in the chicken flock** | | | | | | |
| No | 29 | 94 | Ref. | | Ref | |
| Yes | 61 | 8 | 24.72 (5.59–109.3) | <0.001 | 28.47 (4.97–162.97) | <0.001 |
| **Received training on chicken production** | | | | | | |
| Yes | 6 | 12 | Ref. | | | |
| No | 84 | 90 | 1.87 (0.77–4.53) | 0.168 | | |
| **Knowledge on the purpose of antibiotic use** | | | | | | |
| Used to treat viral diseases | 10 | 23 | Ref. | | | |
| Used to treat bacterial diseases | 29 | 42 | 1.59 (0.8–3.14) | 0.183 | | |
| Used to treat all diseases | 50 | 36 | 3.19 (0.79–12.91) | 0.103 | | |
| Used to increase production | 1 | 1 | 2.3 (0.89–5.87) | 0.082 | | |
| **Farmers education** | | | | | | |
| Graduate | 9 | 17 | Ref. | | | |
| Higher Secondary | 8 | 17 | 0.88 (0.34–2.35) | 0.812 | | |
| Secondary | 44 | 44 | 1.89 (0.73–4.89) | 0.19 | | |
| Primary | 23 | 21 | 2.07 (0.88–4.84) | 0.094 | | |
| Illiterate | 6 | 3 | 3.78 (0.23–61.6) | 0.351 | | |
| **Farming experiences** | | | | | | |
| > 10 years | 26 | 47 | Ref. | | | |
| 5–10 years | 23 | 25 | 1.66 (0.74–3.75) | 0.22 | | |
| 1–5 years | 41 | 30 | 2.47 (1.31–4.67) | 0.005 | | |
| **Heard of AMR** | | | | | | |
| Yes | 45 | 68 | Ref. | | | |
| No | 45 | 34 | 2 (0.77–5.22) | 0.157 | | |
| **Antibiotic suggested by** | | | | | | |
| Veterinary doctor | 61 | 71 | Ref. | | | |
| Feed dealer | 27 | 27 | 1.16 (0.47–2.86) | 0.741 | | |
| Farmer | 2 | 4 | 0.58 (0.19–1.76) | 0.338 | | |

Model fit: Model χ2 6.86, p 0.1432and $R^2$0.3835.

Sonali farms (aOR 28.47, 4.97–162.97). The odds were also higher for broiler (aOR 2.33, 1.02–5.33) and layer (aOR 3.79, 1.71–8.38) farms located in Gazipur than Cumilla district, as well as for broiler farms for which the farmers had inappropriate knowledge on antibiotic use (aOR 3.25, 1.69–6.26) (Tables 8 and 9). The final model selected for layer ($\chi$2 10.67, $p$ = 0.64, and $R^2$ = 0.33) and Sonali ($\chi$2 6.86, $p$ = 0.14, and $R^2$ = 0.38) seemed to fit the data well, whereas the model selected for broiler ($\chi$2 20.81, $p$ = 0.06, and $R^2$ = 0.39) did not fit as well (Tables 8–10).

## Discussion

This study surveyed a large number of commercial chicken farms from a wide range of geographical locations in Bangladesh, including different chicken production types and scales (small to large). This study showed that commercial chicken farmers frequently administer antibiotics to chickens, in particular broiler and Sonali chickens. Under this study, antibiotic usage data was collected over different timeframes, in the 24 hours, 72 hours and 14 days preceding our farm visit. No previous published studies have reported antibiotic usage in commercial chicken production using similar time frames for Bangladesh. Few previous studies from Bangladesh have estimated the proportion of farms using antibiotics, but accurate comparison between production systems has been precluded in the absence of detailed timeframes. It has been reported that 54%-100% of broiler and layer farms administer antibiotics from the start of the production cycle to the day they were surveyed [15,21,22]. Antibiotic usage in commercial poultry in many low- and middle-income countries including Sudan, Tanzania, Vietnam, Philippines, Pakistan, Nepal, Ghana, Nigeria and Cameroon varied from 44–100%, either at the time of farm visits or during the chicken production cycle [5–7,23–28]. For comparison, the proportion of broiler chicken farms using antimicrobials was 26% on day 1 and 49% within the first week of production in nine European countries [29].

According to the World Organization for Animal Health (OIE) list of antimicrobial agents of veterinary importance, many of the antibiotics reported to have been used in recruited farms (including doxycycline, oxytetracycline, amoxicillin, neomycin, erythromycin, tylosin, ciprofloxacin, enrofloxacin) are considered as Veterinary Critically Important Antimicrobial Agents (VCIA). Among these VCIA, fluoroquinolones and third and fourth generation of cephalosporin are considered to be critically important for both human and animal health. Colistin is recognized as category of Highest Priority Critically Important Antimicrobials by WHO [30]. Previous studies conducted in Bangladesh, Pakistan, Vietnam, Philippines, Tanzania, Pakistan, Ghana, Nigeria and Cameroon have also reported common usage of such antibiotics of critical importance for animal and/or human health in commercial broiler and layer chickens [5,6,9,15,24,25,27,31,32].The usage of antibiotics in animal production systems is a global issue. Some antibiotics (colistin, fluoroquinolones and third- and fourth-generation cephalosporins) are advised not to be used in food-producing animals [33]. The World Health Organization (WHO) has recommended complete restriction of all classes of medically important antibiotics in food producing animals for prophylactic purposes [33]. Surprisingly, a large proportion of commercial chicken farms from many countries including Bangladesh (17–57%), Pakistan (60%), Thailand (63%) and Vietnam (36%), as well as across nine European countries (18–26%), have reported use of critically important antimicrobial (CIA) classes in commercial poultry production [8,15,26,29,34]. Interestingly, our study also identified a large number broiler, layer and Sonali farms that had used critically important antimicrobial classes such as colistin and fluoroquinolones. This extensive use of medically important antibiotics in commercial chicken production may promote the development of resistance in microbial populations infecting animals and humans.

Recognizing that use of antibiotics for prophylactic and growth promotion purposes in chicken production sectors is a matter of concern worldwide, our study recorded the frequent prophylactic usage of antibiotics in commercial farms. Many farmers reported that they used antibiotics for fattening purposes. Earlier studies from Bangladesh reported similar evidence of antibiotic use for prophylaxis (23–32%) and growth promotion (8%) in commercial chicken production [16,22]. The routine use of antibiotics at different stages of the production cycle in commercial chicken for prophylactic purposes has also been reported in Cameroon (11%), Pakistan (100%), Nigeria (29–60%), Nepal (22%) and Thailand (38%) [5–9,28]. OIE and WHO advise to avoid antimicrobials for prophylactic purposes in the absence of clinical signs in food-producing animals [30,33]. In parallel, the Bangladesh government passed a law in 2010 to ban the introduction of antibiotics into animal feed during manufacturing [17]. However, no guidelines or policies are available regarding the appropriate use of antibiotics in animal production sectors. The regular usage of antibiotics for prophylactic and growth promotion purposes in healthy animals can play a significant role in the emergence of antibiotic resistance [35].

This study identified some factors that were associated with increased antibiotic usage in commercial chicken production systems. Concurrent chicken morbidity and farm location were significantly associated with increased antibiotic usage in commercial chickens. A study from China reported that lower education levels of farmers and lack of formal agricultural training, likely resulting in poor understanding of AMR, were associated with misuse of antibiotics in chicken farms [36]. According to farmers interviewed in this study, chicken illness was frequently occurring during our farm visit. To treat sick chickens, farmers were often advised to use antibiotics. Farms located in Gazipur and Bogura used antibiotics frequently than in other areas. However, reasons for this were not clearly understood. There might be higher prevalence of diseases in these areas due to high density of chicken farms with different level of biosecurity. Therefore, farmers in these areas could use more antibiotics than other areas.

This study revealed that a large proportion of farmers followed the advice of feed dealers about antibiotic usage. A large number of farmers were financially covenanted to feed dealers that may develop dependency of farmers to feed dealers. As feed dealers also sell antibiotics, they may encourage their purchase by farmers for business interest. Although most of the feed dealers were familiar with AMR, they recommended farmers buying antibiotics for their chickens. Antibiotic recommended by unqualified antibiotic providers needs to be controlled to minimize inappropriate use of antibiotics in commercial chicken production sectors.

This cross-sectional survey may have some limitations. We used purposive sampling instead of random sampling to select farms in each sampling stratum. Limited time and funding did not allow us to conduct a census of all farms in each selected upazila to support their random sampling. However, this may not have influenced the study findings as the farm characteristics and reported antimicrobial usage were consistent with an earlier study [15]. The information that we collected from farmers about antibiotic usage may have been affected by social desirability; this may have resulted in an underestimation of the usage of antibiotics.

In conclusion, this cross-sectional survey revealed that the use of antibiotics in commercial chicken production was extensive in Bangladesh. Most antibiotics were administered for therapeutic and prophylactic purposes. Antibiotics were more commonly used in broiler and Sonali than in layer farms. The occurrence of antibiotic use in the 24 hours preceding our visit was significantly higher in flocks with clinically sick chickens than in healthy flocks. The findings from this study emphasize that the improvement of chicken health through good farming practices can help to reduce antibiotic use and the consequential development of antimicrobial resistance. Regular monitoring of antibiotic usage, educating farmers, drug sellers and feed

dealers about effective use of antibiotics, and restricting ease of access to antibiotics, may also be useful to reduce unnecessary use of antibiotics in commercial chicken production systems.

## Supporting information

**S1 File. Questionnaire for AMU data collection in commercial poultry farm.**
(DOCX)

**S2 File. Questionnaire for AMU data collection from animal feed dealer and farmers.**
(DOCX)

## Acknowledgments

We are grateful to chicken farmers and feed dealers for providing their valuable time and information. The icddr,b is also grateful to the governments of Bangladesh, Canada, Sweden, and the United Kingdom for providing core/unrestricted support.

## Author Contributions

**Conceptualization:** Sukanta Chowdhury, Shahana Parveen, Zubair Akhtar, Khaleda Islam, Zobaidul Haque Khan, Mahmudur Rahman, Nitish Debnath, Fahmida Chowdhury.

**Data curation:** Sukanta Chowdhury, Guillaume Fournié, Sumon Ghosh, Md. Ariful Islam, Md. Mahbubur Rashid.

**Formal analysis:** Sukanta Chowdhury, Guillaume Fournié, Damer Blake, Joerg Henning, Patricia Conway, Md. Ahasanul Hoque, Sumon Ghosh, Shahana Parveen, Paritosh Kumar Biswas, Zubair Akhtar, Khaleda Islam, Md. Ariful Islam, Md. Mahbubur Rashid, Ludvic Pelligand, Zobaidul Haque Khan, Mahmudur Rahman, Fiona Tomley, Nitish Debnath, Fahmida Chowdhury.

**Funding acquisition:** Sukanta Chowdhury, Fahmida Chowdhury.

**Investigation:** Sukanta Chowdhury, Sumon Ghosh, Shahana Parveen, Zubair Akhtar, Khaleda Islam, Md. Ariful Islam, Md. Mahbubur Rashid, Zobaidul Haque Khan, Mahmudur Rahman, Nitish Debnath, Fahmida Chowdhury.

**Methodology:** Sukanta Chowdhury, Guillaume Fournié, Damer Blake, Joerg Henning, Md. Ahasanul Hoque, Paritosh Kumar Biswas, Khaleda Islam, Md. Mahbubur Rashid, Ludvic Pelligand, Zobaidul Haque Khan, Mahmudur Rahman, Fiona Tomley, Nitish Debnath, Fahmida Chowdhury.

**Project administration:** Sukanta Chowdhury, Zobaidul Haque Khan, Nitish Debnath, Fahmida Chowdhury.

**Resources:** Sukanta Chowdhury, Nitish Debnath, Fahmida Chowdhury.

**Software:** Sukanta Chowdhury.

**Supervision:** Khaleda Islam, Fiona Tomley, Nitish Debnath, Fahmida Chowdhury.

**Validation:** Sukanta Chowdhury, Guillaume Fournié, Joerg Henning, Patricia Conway, Paritosh Kumar Biswas, Ludvic Pelligand, Mahmudur Rahman, Fiona Tomley, Nitish Debnath, Fahmida Chowdhury.

**Visualization:** Sukanta Chowdhury, Guillaume Fournié, Md. Ahasanul Hoque, Khaleda Islam, Ludvic Pelligand, Fiona Tomley, Fahmida Chowdhury.

**Writing – original draft:** Sukanta Chowdhury.

**Writing – review & editing:** Guillaume Fournié, Damer Blake, Joerg Henning, Patricia Conway, Md. Ahasanul Hoque, Sumon Ghosh, Shahana Parveen, Paritosh Kumar Biswas, Zubair Akhtar, Khaleda Islam, Md. Ariful Islam, Md. Mahbubur Rashid, Ludvic Pelligand, Zobaidul Haque Khan, Mahmudur Rahman, Fiona Tomley, Nitish Debnath, Fahmida Chowdhury.

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
