## [Decision Letter · Decision Letter 0]

8 Aug 2022

PONE-D-22-06825Antibiotic Usage and Its Drivers in Commercial Chicken Production in BangladeshPLOS ONE

Dear Dr. Chowdhury,

Thank you for submitting your manuscript to PLOS ONE. After careful consideration, we feel that it has merit but does not fully meet PLOS ONE’s publication criteria as it currently stands. Therefore, we invite you to submit a revised version of the manuscript that addresses the points raised during the review process.

First and most important is that you give a clear explanation what is different from the former studies you made on the subject. As one of the reviewers noted this publication seems to be the same as 2 former studies. Publishing the same manuscript in different journals cannot be accepted, and will automatically lead to the rejection of the manuscript if not clear difference between this manuscript and the former manuscripts are given. Second the statistical methods used are not adequate and are erroneous so they need through adaptation.. There are thus major issues with the manuscript.Reply also to the other comments and have the manuscript professionally edited for the English language.

We look forward to receiving your revised manuscript.

Kind regards,

Patrick Butaye, DVM, PhD

Academic Editor

PLOS ONE

https://journals.plos.org/plosone/s/file?id=ba62/PLOSOne_formatting_sample_title_authors_affiliations.pdf".

“This study was funded by the Fleming Fund Country Grant to Bangladesh (FF48-416FFCGB1).”

“This study was funded by the Fleming Fund Country Grant to Bangladesh (FF48-416FFCGB1).”

“This study was funded by the Fleming Fund Country Grant to Bangladesh (FF48-416FFCGB1).”

“The authors have no conflict of interest to declare. The funders had no conflict of interest with the study and contents of the manuscript.”

6. We note that Figure 1 in your submission contain [map/satellite] images which may be copyrighted. All PLOS content is published under the Creative Commons Attribution License (CC BY 4.0), which means that the manuscript, images, and Supporting Information files will be freely available online, and any third party is permitted to access, download, copy, distribute, and use these materials in any way, even commercially, with proper attribution. For these reasons, we cannot publish previously copyrighted maps or satellite images created using proprietary data, such as Google software (Google Maps, Street View, and Earth). For more information, see our copyright guidelines: http://journals.plos.org/plosone/s/licenses-and-copyright.

Please upload the completed Content Permission Form or other proof of granted permissions as an """"Other"""" file with your submission.

Natural Earth (public domain): http://www.naturalearthdata.com/.

Reviewers' comments:

Reviewer's Responses to Questions

**Comments to the Author**

1. Is the manuscript technically sound, and do the data support the conclusions?

Reviewer #1: Partly

Reviewer #2: Yes

2. Has the statistical analysis been performed appropriately and rigorously? 

Reviewer #1: No

Reviewer #2: Yes

3. Have the authors made all data underlying the findings in their manuscript fully available?

Reviewer #1: No

Reviewer #2: Yes

4. Is the manuscript presented in an intelligible fashion and written in standard English?

Reviewer #1: No

Reviewer #2: Yes

5. Review Comments to the Author

Reviewer #1: Dear authors,

Thank you for submitting your manuscript on antimicrobial usage in commercial chicken production in Bangladesh.

You collected a lot of data, and the information may be of interest. However, I noted a few flaws in your study design, statistical analyses, and overall, in the way you interpret and draw conclusions on your findings:

- Please provide a summary of the main findings of relevant publications (some of you were co-authors of those recent publications, especially doi: 10.3389/fvets.2020.576113 and https://doi.org/10.3390/vetsci8060111) and how your study is complementary of those publications (at first sight, your study is a repetition of already available information)

- Provide clear objectives and hypotheses for your study

- Statistical analyses are not clear or not adequately presented (ranges, proportions, lack of information on the 95% confidence interval). Some results are presented as significantly significant where they are not. Please use the STROBE Statement—Checklist of items that should be included in reports of cross-sectional studies.

- Both questionnaires (farmers and feed dealers) should be provided for a better understanding of variables and results. Ideally, the database should be available through your publication as well.

- I have noted typos or errors in English (even if English is not my native language) from the first sentence of the abstract (“can contributed”). Please review! Also, I have noted some inhomogeneity in the writing (spacing for example).

- Introduction (context) and discussions should be completely reviewed. The introduction should give to the reader more context on the Bangladesh chicken systems, regulations related to antimicrobial usage, and conclusions of recent studies. The discussion should discuss biases of the study and confounding factors, internal and external validity, and limits.

Please find attached the comments and corrections directly added in your manuscript.

Kind regards.

Reviewer #2: The article is well written and subscribes an very important topic. By publishing this paper, the Bangladesh stakeholders in prescribing antimicrobial will become more aware of the risks of antimicorbial resistance and be more reluctant in prescribing. There are a few things that has to be improved, I've mentioned them in the attached file

6. PLOS authors have the option to publish the peer review history of their article (what does this mean?). If published, this will include your full peer review and any attached files.

Reviewer #1: **Yes: **Hélène Lardé

Reviewer #2: **Yes: **hetty schreurs

---

## [Author Response · Author response to Decision Letter 0]

14 Sep 2022

Dear Dr. Patrick Butaye,

Thank you for sharing the reviewers' helpful comments and suggestions. We have responded

to all comments and made necessary changes in the revised version of the manuscript

uploaded with this submission. Detailed responses to each comment are available in a separate attachment for your consideration.

Sincerely,

Sukanta

---

## [Editor Report · Decision Letter 1]

3 Oct 2022

Antibiotic Usage Practices and Its Drivers in Commercial Chicken Production in Bangladesh

PONE-D-22-06825R1

Dear Dr. Chowdhury,

We’re pleased to inform you that your manuscript has been judged scientifically suitable for publication and will be formally accepted for publication once it meets all outstanding technical requirements.

Kind regards,

Patrick Butaye, DVM, PhD

Academic Editor

PLOS ONE
---

## [Editor Report · Acceptance letter]

7 Oct 2022

PONE-D-22-06825R1 

Antibiotic Usage Practices and Its Drivers in Commercial Chicken Production in Bangladesh 

Dear Dr. Chowdhury:

I'm pleased to inform you that your manuscript has been deemed suitable for publication in PLOS ONE. Congratulations! Your manuscript is now with our production department. 

Kind regards, 

on behalf of

Professor Patrick Butaye 

Academic Editor

PLOS ONE